# Inhibition of Toll-like Receptor 4 Using Small Molecule, TAK-242, Protects Islets from Innate Immune Responses

**DOI:** 10.3390/cells13050416

**Published:** 2024-02-27

**Authors:** Jordan Mattke, Carly M. Darden, Srividya Vasu, Michael C. Lawrence, Jeffrey Kirkland, Robert R. Kane, Bashoo Naziruddin

**Affiliations:** 1Institute of Biomedical Studies, Baylor University, Waco, TX 76706, USA; jordan.mattke@bswhealth.org (J.M.); bob_kane@baylor.edu (R.R.K.); 2Annette C. and Harold C. Simmons Transplant Institute, Baylor University Medical Center, Dallas, TX 75204, USA; carly.darden@bswhealth.org (C.M.D.); jeffrey.kirland@bswhealth.org (J.K.); 3Islet Cell Laboratory, Baylor Scott and White Research Institute, Dallas, TX 75204, USA; vasandhya87@gmail.com (S.V.); michael.lawrence@bswhealth.org (M.C.L.)

**Keywords:** total pancreatectomy with islet autologous transplantation, miRNA, instant blood-mediated inflammatory reaction, TLR4, inflammation

## Abstract

Islet transplantation is a therapeutic option to replace β-cell mass lost during type 1 or type 3c diabetes. Innate immune responses, particularly the instant blood-mediated inflammatory reaction and activation of monocytes, play a major role in the loss of transplanted islet tissue. In this study, we aimed to investigate the inhibition of toll-like receptor 4 (TLR4) on innate inflammatory responses. We first demonstrate a significant loss of graft function shortly after transplant through the assessment of miR-375 and miR-200c in plasma as biomarkers. Using in vitro models, we investigate how targeting TLR4 mitigates islet damage and immune cell activation during the peritransplant period. The results of this study support the application of TAK-242 as a therapeutic agent to reduce inflammatory and innate immune responses to islets immediately following transplantation into the hepatic portal vein. Therefore, TLR4 may serve as a target to improve islet transplant outcomes in the future.

## 1. Introduction

Type 1 diabetes mellitus is a chronic disease characterized by insufficient insulin production following autoimmune destruction of pancreatic β-cells, leading to hyperglycemia [1]. Although this condition is usually managed through the administration of exogenous insulin, one therapeutic option for patients suffering from type 1 diabetes is allogenic islet transplantation to replace the β-cell mass lost to autoimmune destruction [2]. Islet transplantation can also be performed to restore glycemic function following total pancreatectomy [3]. 

Previous reports have demonstrated good glycemic control after islet transplantation to treat type 1 diabetes and the prevention of diabetes following pancreatectomy with improved quality of life. However, many patients require large quantities of islets to achieve complete insulin independence [3,4,5,6]. One of the major hurdles still to overcome in the field of autologous and allogenic islet transplantation is the challenge of innate immune responses immediately following transplantation. It has been estimated that 50% to 70% of islet mass is lost due to this innate immune reaction to islets as they are infused into the portal vein [7]. One of the earliest challenges for transplanted tissue is the instant blood-mediated inflammatory reaction (IBMIR) characterized by platelet consumption and activation of the coagulation and complement pathways, resulting in clot formation around islets as well as immune infiltration immediately following infusion into the portal vein [8]. Effective control of the innate immune response will minimize islet damage and improve islet transplant function.

Toll-like receptor 4 (TLR4) is an innate immune receptor expressed on a variety of cell types within the body. Most notably, TLR4 is known to be stimulated in the presence of bacterial lipopolysaccharide (LPS) as well as damage-associated molecular patterns (DAMPs). Upon stimulation, TLR4 oligomerizes and can signal through the MyD88-dependent and MyD88-independent pathways, resulting in the production of proinflammatory cytokines and the recruitment of leukocytes [9].

In monocytes, stimulation of TLR4 using LPS produces a more inflammatory M1 macrophage phenotype [10]. This is significant in that it has been noted that M1 macrophages are one of the first infiltrating leukocytes into transplanted tissue following islet transplant in a mouse model [11]. A study has also shown that inhibition of TLR4 using siRNA leads to lowered chemotactic and phagocytic activity as well as reduced production of IL-1β and IL-6 in recipients [12]. Inhibition of TLR4 using peptide results in reduced IL-1β, TNF-α, iNOS, and IL-6 in Raw264.7 macrophages exposed to LPS and results in reduced macrophage infiltration in islet grafts [13]. Stimulation of TLR4 within dendritic cells leads to dendritic cell maturation and increased T cell priming [14]. TLR4 also has a significant role in the process of cross-presentation within dendritic cells, as TLR4 stimulation engages Rab34-dependent reorganization of lysosomes and delays antigen degradation as well as increasing Rab14 activity, leading to increased anterograde transport resulting in altered antigen presentation and activation of T cells [15,16]. 

Previous studies in mice have shown that TLR4-deficient allogenic islets display improved survival when transplanted into BALB/c recipients [17]. An in vitro blockade of TLR4 using TLR4 antibody resulted in reduced β-cell apoptosis and T cell activation, and proliferation against allogenic islets as well as indefinite allogenic islet graft survival, although tolerance was not achieved in this study [18]. These results were further confirmed by another study that showed that deficiency in TLR4 in donor islets or blockade of the TLR4 ligand HMGB1 resulted in prolonged graft survival [19]. 

TAK-242 is a small molecule shown to be highly specific for TLR4, with little off-target effects, and has been used in clinical trials for the treatment of severe sepsis [20,21]. Since previous work has demonstrated TLR4 having a substantial role in islet transplantation leading to early graft failure, our laboratory sought to investigate whether inhibition of TLR4 signaling using TAK-242 could lead to more favorable transplant outcomes [22]. Results of the study by Chang et. al. revealed that islets transplanted under the kidney capsule with soluble TAK-242, or islets surface-modified with an NHS-PEG linker connected to a releasable TAK-242, showed improved islet recovery when compared to mice treated with control islets [23].

In this study, we will show that inhibition of TLR4 using TAK-242 reduces innate immune responses such as IBMIR and leukocyte activation, leading to attenuation of islet stress and damage. Our study uses miR-375 and miR-200c as biomarkers to assess islet damage in both clinical samples and cell culture experiments. We also report that TAK-242 inhibits the activation of inhibition of innate immune cells, which then reduces the proliferation and activation of CD8+ T cells. Overall, our results reveal a broader role for TLR4 in the protection of islet transplants from innate immune responses.

## 2. Materials and Methods

### 2.1. Patients and Sample Collection

All the patients included in this study underwent total pancreatectomy with islet autologous transplantation (TPIAT) at Baylor University Medical Center in Dallas, TX, USA. To validate miR-375 and miR-200c as reliable biomarkers to assess islet damage, first, we analyzed plasma samples collected from TPIAT patients during the time of islet infusion. Previously published work showed that optimal outcomes following TPIAT occur when patients receive >5000 IEQ/kg [24]. We therefore narrowed our focus to patients receiving >5000 IEQ/kg during TPIAT. Blood samples were collected prior to islet infusion and 1 h after the completion of infusion. Three-month follow-up data were collected for each patient in this study. Table 1 presents the patient and pancreas characteristics.

### 2.2. Islet Isolation

Islet isolations were carried out at the Baylor Scott and White Research Institute from deceased donor and chronic pancreatitis pancreases. Islet isolations were performed according to previously established practices for clinical transplantation [25]. The pancreases were decontaminated in an antibiotic before being perfused with collagenase enzyme (Vitacyte, Indianapolis, IN, USA) solution, followed by mechanical digestion in a Ricordi chamber and purification in a density gradient solution [25]. The viability of the islets was confirmed using fluorescein diacetate and propidium iodide staining. The islets were allowed to recover overnight in PIM(S) media supplemented with PIM(ABS) Human AB Serum Supplement and PIM(G) Glutamine and glutathione supplement (Prodo Laboratories Inc., Aliso Viejo, CA, USA).

### 2.3. Surface Modification of Islets

The islet surfaces were modified with TAK-242 using copper-free click chemistry, previously carried out by Chang et al. [23]. The islets were suspended in Kreb’s Ringer bicarbonate buffer containing 5.6 mM of glucose and 20 μM of NHS-Linker-TAK-242 dissolved in DMSO for 30 min. Control islets were incubated in Kreb’s Ringer bicarbonate buffer containing an equivalent amount of DMSO as NHS-Linker-TAK. The islets were then washed and suspended in RPMI 1640 with 10% fetal bovine serum (FBS) and Anti/Anti prior to beginning experiments.

### 2.4. In Vitro IBMIR

After 24 h of culture, 500 IEQ of islets was administered into heparinized tubes. The control samples were mixed with 500 μL of RPMI with 10% FBS. For IBMIR simulation, the islets were mixed with 500 μL of whole allogenic blood. The tubes were then incubated for 3 h at 37 °C with agitation. For the samples treated with soluble TAK-242, TAK-242 was administered to a final concentration of 10 μM. The concentration of 10 μM was determined in order to keep consistency between this experiment and all other assays, which showed significant changes in immune responses starting at 10 μM. All the other samples contained an equivalent amount of DMSO as TAK-242-treated samples. At the beginning of the experiment and after 3 h of incubation with agitation, all the samples were centrifuged, and plasma was collected for miRNA analysis.

### 2.5. Clotting Assay

An amount of 50 IEQ of islets was suspended in 100 μL of platelet-poor plasma in the presence and absence of 10 μM of TAK-242 in a 96-well plate. Plasma without islets was added to wells to serve as a negative control. Then, 25 mM CaCl_2_ was added to each well of the plate before the plate was transferred to a Cytation 5 (BioTek, Winooski, VT, USA) to measure the absorbance every minute at 405 nm.

### 2.6. One-Way Mixed Lymphocyte Reaction

Human peripheral blood mononuclear cells (PBMCs) were isolated by Ficoll Paque gradient centrifugation. The PBMCs were labeled with CFSE cell proliferation dye (ThermoFisher Scientific, Waltham, MA, USA). A total of 1 × 10^5^ PBMCs were mixed with 1 × 10^5^ mitomycin C-treated human splenocytes labeled with a CellTrace Violet Cell Proliferation Kit (ThermoFisher Scientific, Waltham, MA, USA) in a U-bottom 96-well plate with 55 μM of IL-2 and 5 μg/mL of LPS.

### 2.7. CD8+ T Cell Activation Assay

Following the isolation of PBMCs, the T cells were enriched with an EasySep Human T Cell Enrichment Kit (STEMCELL Technologies, Vancouver, BC, Canada). CD8+ T cells were then selected using an EasySep Human CD4 Positive Selection Kit II (STEMCELL Technologies, Vancouver, BC, Canada). The cells were then labeled with CFSE or a CellTrace Violet Proliferation Kit. The CD8+ T cells were then mixed with Dynbeads Human T-Activator CD3/CD28 (ThermoFisher Scientific, Waltham, MA, USA) at a 1:1 ratio in a U-bottom 96-well plate. The cells were monitored for activation and proliferation by flow cytometry.

### 2.8. In Vitro Co-Culture of Allogenic PBMC and Islets

PBMCs were isolated by Ficoll Paque gradient centrifugation. The PBMCs were labeled with a CellTrace Violet Cell Proliferation Kit (ThermoFisher Scientific, Waltham, MA, USA). A total of 1 × 10^5^ PBMCs were then mixed with ~15 IEQ of hand-picked islets in RPMI medium supplemented with 10% FBS, Anti/Anti (Gibco, Aliso Viejo, CA, USA), and 55 μM IL-2 in a U-bottom 96-well plate.

### 2.9. Cytokine Quantification

After 24 h, of co-culture supernatant from each well was removed. The supernatant was then analyzed for the quantities of IL-1β, IL-6, IL-8, IP10, MCP-1, and TNFα by a Milliplex Human Cytokine/Chemokine/Growth Factor Panel A (MilliporeSigma, Burlington, MA, USA) according to the manufacturer’s protocol.

### 2.10. RT-qPCR for Secreted Stress and Damage miRNA

Following the collection of plasma and cell culture supernatant, miRNA was isolated using a miRNeasy Serum/Plasma Advanced Kit (Qiagen, Hilden, Germany). A UniSp6 synthetic spike was added prior to column isolation. The miRNA was reverse-transcribed using a miRCURY LNA RT Kit (Qiagen, Hilden, Germany). qPCR was performed using a miRCURY LNA SYBR Green PCR Kit (Qiagen, Hilden, Germany) using primers for miR-375 and miR-200c (Qiagen, Hilden, Germany).

### 2.11. Flow Cytometry

The islets were removed from each co-culture by hand-picking. The cells were then administered into 5 mL polystyrene tubes through 70 μM filters. The T cells were stained with anti-human CD3 Alexa Fluor 700, anti-human CD4 APC, anti-human CD8 APC-Cy7, anti-human CD69 PE, and anti-human CD25 BV786 (Biolegend, San Diego, CA, USA). The dendritic cells were stained with anti-human CD14 Brilliant Violet 605, anti-human CD11c PE-Cy7, anti-human CD80 BUV737, and anti-human CD83 BV785 (Biolegend, CA, USA). The cells were then analyzed on a BD LSRFortessa Cell Analyzer.

### 2.12. Statistical Analysis

The different groups were compared by unpaired two-tailed Student’s t-test. The flow cytometry data were analyzed using FlowJo 10.8.1. All the data were analyzed using GraphPad Prism 9.1.2 (GraphPad Software, La Jolla, CA, USA). Our clinical data sample size was confirmed appropriate using power calculation, which used 0.8 for the correlation coefficient, 0.05 for the alpha value, and 0.8 for the power value.

## 3. Results

### 3.1. Elevated miR-375 and miR-200c Correspond to Poor Islet Function following TPIAT

First, we wanted to identify biomarkers to assess islet damage. Previously, we had shown that intraportal infusion of islets into patients immediately results in islet damage, as evidenced by elevated microRNA levels in the plasma [26]. We analyzed plasma samples from TPIAT patients who received an islet dose of >5000 IEQ/kg to reliably assess islet damage. We observed a several-fold increase in both miR-375 and miR-200c 1 h after islet infusion when compared to pretransplant samples. When comparing the fold change of miR-375 and miR-200c to transplant outcomes 3 months after surgery, several trends were observed. First, we demonstrated that both miR-375 and miR-200c correlate strongly with HbA1c levels at 3 months following TPIAT (Figure 1). The miR-375 and miR-200c also had a positive correlation to insulin usage 3 months following TPIAT (*p* < 0.05) (Figure 1), indicating that miR-375 and miR-200c can be used as biomarkers to assess islet damage and also to predict islet function following transplantation.

### 3.2. TAK-242 Reduces Damage to Islets Exposed to IBMIR In Vitro

To evaluate the protective effects of TAK-242 on islet damage, we created an in vitro model of IBMIR and measured the miR-375 and miR-200c release. When the islets were exposed to whole blood, there was a significant increase in miR-375 (control islet = 6.017 ± 0.177 FC vs. control IBMIR = 12.294 ± 1.448 FC, *p* < 0.001) and miR-200c (control islet = 6.327 ± 0.761 FC vs. control IBMIR = 33.273 ± 0.666 FC, *p* < 0.001) in response to the presence of blood, which was not noted in samples that did not contain a mixture of blood and islets (Figure 2). However, when TAK-242 was present in soluble form or released locally from the surface of islets, the islets secreted less miR-375 (control IBMIR = 12.294 ± 1.448 FC, MAP84 IBMIR = 12.154 ± 0.411 FC, TAK-242 IBMIR = 8.149 ± 0.375 FC, modified IBMIR = 8.157 ± 1.102 FC, *p* < 0.05), and miR-200c (control IBMIR = 33.273 ± 0.666 FC, MAP84 IBMIR = 31.674 ± 1.990 FC, TAK-242 IBMIR = 17.665 ± 0.35 FC, modified IBMIR = 19.761 ± 3.26 FC, *p* < 0.05) when compared to untreated islets or islets treated with inactive TAK-242 analog, MAP84 (Figure 2). The decrease in these miRNAs can be taken as a sign that the islets are undergoing less stress and damage due to the IBMIR reaction in the presence of TAK-242 [27]. In order to assess whether this reduction in damage was due to reduced immune cell activation or reduced complement and coagulation cascades, we performed a clotting assay using islets and platelet-poor plasma to evaluate whether TAK-242 affects the clotting process. In this experiment, we did not note a significant reduction in clotting times in mixtures with TAK-242 compared to clotting times with no treatment. Therefore, the remainder of our study focused on inflammation and damage due to immune cell infiltration and activation.

### 3.3. TAK-242 Inhibits T Cell Activation and Proliferation in a One-Way Mixed Lymphocyte Reaction with Splenocytes

Because β-cells of islets are able to secrete their own cytokines, termed “isletokines”, we assessed whether the TAK-242 was affecting the isletokine secretion from islets, or whether TAK-242 was directly inhibiting innate immune responses [28]. To do this, we conducted a one-way mixed lymphocyte reaction using human allogenic PBMCs and mitomycin C-treated splenocytes in the presence of supplemental IL-2 to assist with T cell proliferation, and LPS to assist with the activation of dendritic cells responsible for presenting antigen to and priming T cells. To begin, we first observed the activation markers CD83 and CD80 on dendritic cells. Following an overnight incubation of PBMCs and splenocytes, the LPS-treated samples showed a remarkable increase in CD83 (control = 2947 ± 256, LPS control = 4672 ± 64, LPS MAP84 = 5679 ± 331, *p* < 0.005) and CD80 expression (control = 2011 ± 364, LPS control = 15,691 ± 463, LPS MAP84 = 12,054 ± 631, *p* < 0.05) in the LPS control samples and LPS MAP84 samples (Figure 3). However, this increase in CD83 (LPS TAK-242 = 2864 ± 99, *p* < 0.0005) and CD80 (LPS TAK-242 = 653 ± 106, *p* < 0.001) was not noted in the LPS TAK-242-treated samples, indicating a reduction in the activation of dendritic cells in these samples.

We next analyzed the supernatant from this mixed lymphocyte reaction using a Luminex assay. In this assay, we observed that administration of the TAK-242 ligand, LPS, resulted in significantly increased expression of IL-1β (control = 12.98 ± 16.83 pg/mL, LPS control = 4072 ± 625 pg/mL, LPS TAK-242 = 0 ± 0 pg/mL, *p* < 0.0001), IL-6 (control = 114 ± 107 pg/mL, LPS control = 11,506 ± 1409 pg/mL, LPS TAK-242 = 13.1 ± 1.6 pg/mL, *p* < 0.0001), IL-8 (control = 7866 ± 2071 pg/mL, LPS control = 53,984 ± 3944 pg/mL, LPS TAK-242 = 5297 ± 1144 pg/mL, *p* < 0.0001), and TNFα (control = 446 ± 277 pg/mL, LPS control = 8330 ± 2214 pg/mL, LPS TAK-242 = 111 ± 36 pg/mL, *p* < 0.005). The expression of these cytokines in the supernatant was significantly inhibited by the presence of TAK-242 (Figure 3). Although MCP-1 (control = 1513 ± 665 pg/mL, LPS control = 1380 ± 171 pg/mL, LPS TAK-242 = 182 ± 29 pg/mL, *p* < 0.0005) and IP-10 (control = 451 ± 60 pg/mL, LPS control = 460 ± 50 pg/mL, LPS TAK-242 = 338 ± 13 pg/mL, *p* < 0.05) were not increased by LPS, the addition of TAK-242 significantly reduced the expression of each of these cytokines (Figure 3).

### 3.4. TAK-242 Directly Inhibits the Activation of CD8+ T Cells

We next investigated whether administration of TAK-242 resulted in decreased priming and activation of T cells in the mixed lymphocyte reaction. The initial results showed that TAK-242 decreased the proliferation of CD8+ T cells in a dose-dependent manner (Figure 4). Our observations indicated that combining allogenic splenocytes with LPS as an adjuvant resulted in a synergistic effect leading to an increased number of CD8+ CD69+ CD25+ T cells. As CD8+ T cells are known to express TLR4 and react to markers of sterile inflammation such as IL-12, we next investigated whether TAK-242 directly inhibits CD8+ T cell activation [29]. Upon stimulation with CD3/CD28 antibody-coated beads, we noted a dose-dependent decrease in CD8+ T cell activation, as well as proliferation in the presence of TAK-242 that was not affected by inactive analog, MAP84 (Figure 5). We also confirmed the presence of TLR4 on these cells and noted increased TLR4 expression as these cells became activated (Figure 5).

### 3.5. T Cell Activation and Islet Damage Is Inhibited by the Presence of TAK-242

The mixed lymphocyte reaction was repeated utilizing islets instead of splenocytes. Soluble TAK-242 as well as islets modified with NHS-Linker-TAK-242 showed significantly fewer CD69+ CD8+ T cells (control = 47.8 ± 3.2%, TAK-242 = 35.35 ± 1.35%, MAP84 = 49.35 ± 7.15%, TAK-242 modified = 32 ± 0.2, *p* < 0.05) after 24 h of co-culture (Figure 6). This decreased activation was accompanied by decreased stress and damage to miRNA-375 (control = 6.60 ± 1.13 fM, TAK-242 = 0.927 ± 0.170 fM, MAP84 = 10.413 ± 4.881 fM, TAK-242 modified = 0.258 ± 0.0570 fM, *p* < 0.05) and miRNA-200c (control = 2.266 ± 0.307f, TAK-242 = 0.148 ± 0.0634 fM, MAP84 = 2.111 ± 0.232 fM, *p* < 0.05) (Figure 6). There was also a significant decrease in IL-1β (control = 1310 ± 32 pg/mL, TAK-242 = 37 ± 2.9 pg/mL, MAP84 = 1090 ± 45.5 pg/mL, *p* < 0.05), IL-6 (control = 11,792 ± 81 pg/mL, TAK-242 = 5592 ± 825 pg/mL, MAP84 = 12,216 ± 419 pg/mL, *p* < 0.05), and TNFα (control = 1558 ± 254 pg/mL, TAK-242 = 649 ± 172 pg/mL, MAP84 = 1298 ± 211 pg/mL, *p* < 0.05) present in the supernatant of the TAK-242-treated samples (Figure 6). Islets extracted from control and TAK-242-treated co-cultures showed improved islet structure in the cultures containing TAK-242.

## 4. Discussion

In this study we have shown that inhibition of TLR4 using a small molecule, TAK-242, attenuates damage to islets due to IBMIR and other innate immune reactions. Based on previously published research, it is known that islets release increased amounts of miRNAs in response to stress and damage. Our previously published work demonstrated that miR-375 and miR-200c are among the most abundant islet-specific exosomal miRNAs when islets are subjected to cytokine and hypoxic stress as the islet viability decreases [26]. Using this information, we first sought to investigate whether these miRNAs can be used as diagnostic biomarkers to assess islet graft damage and functional outcomes. Using plasma samples obtained prior to and after islet infusion during TPIAT, we determined that increases in miR-375 and miR-200c were correlated to increased HbA1c and increased use of insulin 3 months after transplant, suggesting that early damage to the autologous islet grafts results in poor functional outcomes in TPIAT. We then used these markers in an in vitro model of IBMIR in order to evaluate whether TAK-242 could lessen the damage to islets as measured by the release of these biomarkers. We observed that there was significantly reduced miRNA-375 and miRNA-200c in mixtures of whole allogenic blood in the presence of TAK-242 than in control conditions or in the presence of the inactive TAK-242 analog, MAP84 (Figure 1). Although miR-375 and miR-200c have proven to be very sensitive biomarkers for assessing islet damage, these markers do not provide real-time results of islet stress and damage as sample processing takes time, and these markers are differentially expressed after islet damage has occurred. A previous study carried out focusing on the inhibition of NF-KB showed that inhibition of NF-κB was able to attenuate IBMIR in the form of increased viability, decreased cytokine expression, and decreased neutrophil infiltration [27]. Because one of the major targets of TLR4 is NF-κB, and TLR4 is one of the major receptors for DAMPs, it can be assumed that TLR4 can play a role in the inflammation and infiltration of leukocytes associated with IBMIR.

Three major components of an IBMIR reaction are coagulation, complement activation (mostly in allogenic and xenogenic models), and activation of inflammatory response. It has been shown that targeting coagulation and complement activation with dextran sulfate results in improved islet transplant outcomes in an in vitro model and nonhuman primate in vivo model [30]. Our results show that TAK242 does not inhibit the coagulation (Figure 1c). Complement activation has been shown to trigger TLR4 signaling, mainly through C5a, resulting in increased NF-κB activity and cytokine production in monocytes [31]. Furthermore, C5a addition to neutrophils showed an upregulation of CD14 and TLR4-activated mRNAs such as IL-8 [32]. However, TLR4 has not been shown to influence the activation of complement pathways. Islets are known to express TLR4, and future studies should investigate whether TAK-242 can mitigate the damage to islets caused by complement activation.

Previous work carried out in our laboratory validated that TAK-242 treatment to islets, as well as modifying the surface of islets with TAK-242, does not affect glucose-stimulated insulin secretion or viability [23,25]. Further, in a syngeneic diabetic mouse model of islet transplantation, administration of soluble TAK-242 or use of TAK-242 surface-conjugated islets significantly improved the achievement of normoglycemia [25]. This same study also evaluated the use of TAK-242 in a syngeneic model of islet transplant resulting in significantly improved transplant outcomes for islets surface-modified with TAK-242 [23]. We do acknowledge that this transplant model was carried out in the kidney capsule of mice where there is less exposure to immune and circulatory factors, which contribute to islet loss. Therefore, future experiments utilizing TAK-242 should focus on preclinical in vivo experiments utilizing the hepatic portal vein for islet infusion in order to assess the effect of TAK-242 on long-term islet transplant outcomes.

After showing that the presence of TAK-242 resulted in no significant changes in clotting times following the mixture of platelet-poor plasma and islets (Figure 2c.), we next focused on the activation of innate immune cells. As the surface modification strategy of islets utilized copper-free click chemistry with a short-chain PEG, which is smaller than proteins, lipids, and carbohydrates of receptors on the cell surface, the surface modification strategy would not prevent receptor or complement binding to the islet surface. Therefore, the most drastic effects of our surface modification would be the effects of TAK-242 following β elimination and release from the islet surface. We then demonstrated that the presence of TAK-242 inhibits immune responses in a one-way mixed lymphocyte reaction experiment, as dendritic cell activation was decreased in the presence of TAK-242 (Figure 2). This shows that not only could TLR4 stimulation affect transplanted donor islet tissue, but it could inhibit the development of recipient immune responses as well. Antigen presentation and dendritic cell cross-dressing play a major role in the activation of T cells following transplantation, leading to rejection [30,31]. TLR4 plays a significant role in the process of antigen processing and presentation within monocytes, as it has been shown that TLR4 stimulation leads to decreased antigen degradation as well as increased anterograde transport of endosomes [15,16]. For this reason, future studies should investigate whether donor antigen presentation is inhibited in monocytes in the presence of TAK-242, which could be a possible reason for decreased CD8+ T cell activation.

This study has also shown that inhibition of TLR4 using TAK-242 results in significantly lower activation of CD8+ T cells (Figure 5). To validate this finding, we decided to test whether TAK-242 specifically inhibits antigen presentation to T cells, or whether TAK-242 is able to directly inhibit T cell activation. We tested this by using polyclonal stimulation of CD8+ T cells with CD3/CD28 antibody-coated beads. In this experiment, we noted a dose-dependent decrease in T cell activation as well as a proliferation in the response to TAK-242. We also noticed an increase in TLR4 expression on these cells, as they became activated and proliferated (Figure 5). Previous research has confirmed the presence and responsiveness of TLR4 to LPS [32]. However, to our knowledge, this is the first report in which inhibition of TLR4 was shown to interfere with TCR signaling. Therefore, a focus of future research should be to investigate how TCR signaling is supported by TLR4 in the CD8+ T cells.

Previous work has shown that TAK-242 can decrease proinflammatory cytokine production in LPS-stimulated macrophages [33]. TAK-242 has also been shown to decrease neutrophil extracellular trap production by neutrophils during acute rejection of liver transplants [34]. In our study, we demonstrate that TAK-242 decreases the production of IL-1β, IL-6, IL-8, IP-10, MCP-1, and TNFα in an allogenic co-culture of PBMCs and splenocytes (Figure 3). The presence of these cytokines in the supernatant of these reactions can be taken as an indirect sign of reduced macrophage and neutrophil infiltration and activation. It is well known that activated macrophages produce IL-1β, IL-6, IL-8, and TNFα, which further contribute to immune infiltration and activation [35]. Neutrophil migration is also strongly influenced by the presence of IL-8 [36]. MCP-1 is recognized as one of the key chemokines regulating macrophage and monocyte recruitment and infiltration [37]. IL-1β has been found to be a key marker of β-cell damage and death in islet transplantation [38]. IL-6, IL-8, and IP-10 were all found to be increased immediately following transplant in human autologous islet transplants into the portal vein [39]. Blockade of IL-6 and TNFα resulted in reduced production of IL-6, IL-8, and MCP-1, with improvement in islet function following transplant [40]. Although it has been shown that β-cells are capable of secreting IP-10, our observation of a one-way mixed lymphocyte reaction showed that responder cells are also capable of producing IP-10 [28]. This expression of IP-10 can be inhibited through the inhibition of TLR4 with TAK-242. Previous research has shown that blockade of IP-10 in a mouse islet transplant model resulted in significantly less lymphocytic infiltrate and longer graft survival compared to wild type, showing that inhibition of TLR4 is an indirect mechanism of attenuating IP-10 production and immune infiltration to transplanted islet tissue [41]. Low secretion of MCP-1 was found to be the most relevant factor of long-lasting insulin independence in islet transplantation and is produced by islets even in the absence of inflammation [42]. Although stimulation of TLR4 using LPS did not lead to statistically significant increases in either IP10 or MCP-1, the presence of TAK-242 significantly decreased the production of these cytokines in these one-way mixed lymphocyte reactions, showing that TLR4 plays a supporting role in the production of these cytokines. Therefore, we believe that inhibition of TLR4 affecting the production of all of these cytokines will result in improved outcomes in allogenic islet transplantation through modulation of islet inflammation as well as innate immune reactions. This hypothesis is supported by representative images showing improved islet structure following the co-culture of islets with PBMCs (Figure 6f,g).

As the IBMIR reaction is a multifaceted response with more than a single cause of inflammation, future studies may focus on the other mechanisms of TLR4 inhibition and how that contributes to the overall inflammation response to islets and IBMIR. TLR4 is expressed on immune cells not covered in this paper, such as neutrophils, B cells, and macrophages, which can contribute to the loss of islet grafts. Figure 7 shows the effects of TAK-242 inhibition of TLR4 signaling and its modulation of innate immune responses in the context of transplants. TLR4 also plays a significant role in the activation of platelets and their interactions with neutrophils to produce neutrophil extracellular traps [43]. Therefore, future studies could also focus on these diverse populations to investigate how targeting TLR4 could attenuate inflammation within these cells and particles. We also acknowledge a shortcoming of this study was in failing to investigate specific T cell populations such as naïve, memory, effector, and regulatory T cells, which play a significant role in the long-term success or failure of islet transplants. As the focus of this study was on innate responses following transplant, our focus was on short-term activation markers on monocytes and T cells. Future work investigating long-term transplant outcomes with TAK-242 should focus on altered T cell phenotypes following transplant. We would also like to point out the limitations of this study, such as the limited availability of qualified patient samples. Although we did meet the minimum of 10 patient samples computed in the power calculation, a larger patient cohort would have strengthened the results of our experiment. Our results are also limited in that we did not obtain histological images of islets following exposure to blood with and without TAK-242, which could provide additional insight into the infiltration of macrophages and neutrophils. As much of the work carried out focused on in vitro experiments, the next logical step for future studies should be to apply TAK-242 to an allogenic islet transplant model in mice to examine the effects of TAK-242 on allogenic islet transplant outcomes in the portal vein.

## 5. Conclusions

Taken together, these results demonstrate TAK-242 as a viable therapeutic to help attenuate early innate graft damage due to host innate immune responses. As previous research has shown that inhibition of TLR4 can lead to prolonged graft survival but not tolerance, future research should utilize not only TAK-242 but also compounds such as rapamycin, shown to enhance Treg activity in order to promote complete graft acceptance and long-term glycemic control in allogenic islet transplantation [44].

## Figures and Tables

**Figure 1 cells-13-00416-f001:**
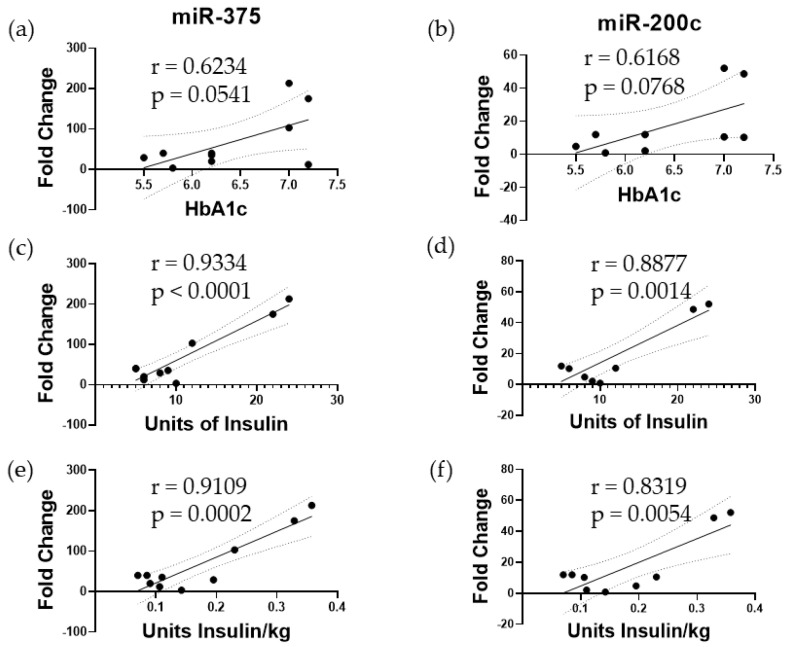
Correlation analysis of miRNA expression 1 h post-infusion with 3-month transplant outcomes. The fold change in (**a**) miR-375 and (**b**) miR-200c showed a positive trend when compared to 3-month hemoglobin A1c values. (**c**) miR-375 and (**d**) miR-200c fold change also showed a significant positive trend when compared to insulin use at 3 months post-TPIAT. This same trend was observed when (**e**) miR-375 and (**f**) miR-200c were compared to the insulin dose 3 months after TPIAT.

**Figure 2 cells-13-00416-f002:**
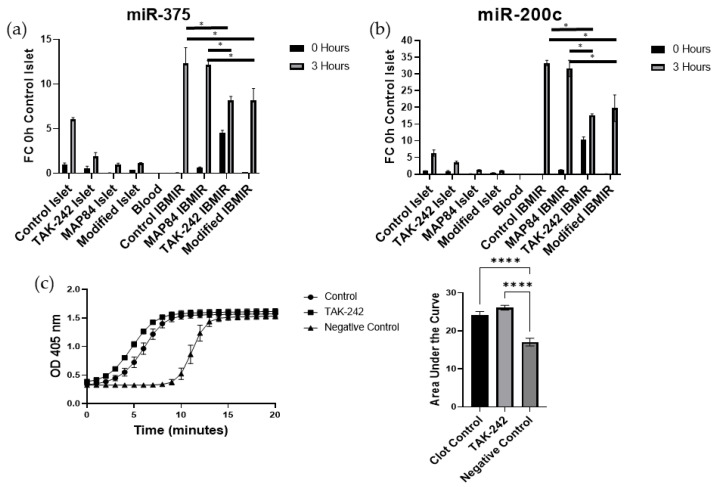
Inhibition of islet damage by TAK-242 in vitro as measured by the expression of (**a**) miR-375 and (**b**) miR-200c during an in vitro IBMIR reaction. Aliquots of islets and blood were mixed in vitro in the presence of TAK-242 or structural analog, MAP84, and the supernatant was analyzed for miR-375 and miR-200c using RT-PCR; (**c**) clotting assay results for islets exposed to platelet-poor plasma. * *p* < 0.05, **** *p* < 0.0001.

**Figure 3 cells-13-00416-f003:**
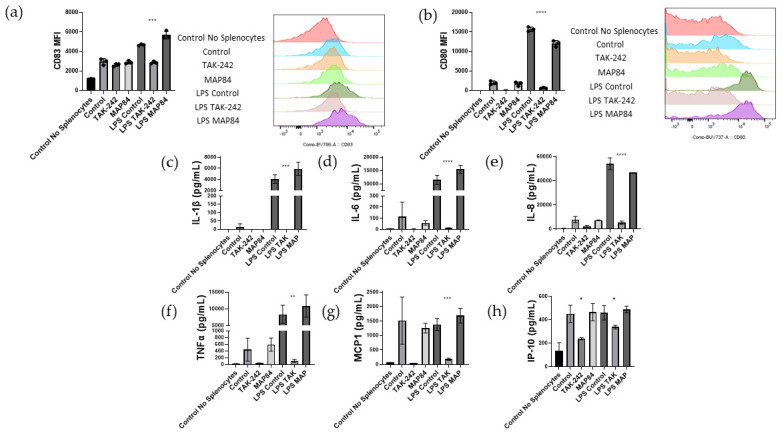
(**a**) CD83 and (**b**) CD80 expression was measured on dendritic cells 24 h after PBMCs were mixed with allogenic splenocytes. Cytokines (**c**) IL-1β, (**d**) IL-6, (**e**) IL-8, (**f**) TNFα, (**g**) MCP1, and (**h**) IP-10 were quantified in the supernatant of this one-way mixed lymphocyte reaction. (* *p* < 0.05, ** *p* < 0.005, *** *p* < 0.0005, **** *p* < 0.0001).

**Figure 4 cells-13-00416-f004:**
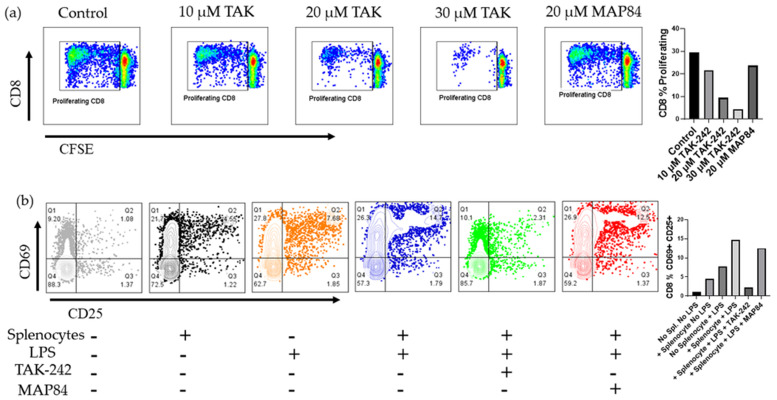
(**a**) Proliferation of CD8+ T cells in a one-way mixed lymphocyte reaction of allogenic PBMCs and mitomycin C-treated splenocytes. (**b**) CD69 and CD25 expression was assessed on CD8+ T cells in this one-way mixed lymphocyte reaction, which also utilized LPS as an adjuvant to enhance immune cell responses.

**Figure 5 cells-13-00416-f005:**
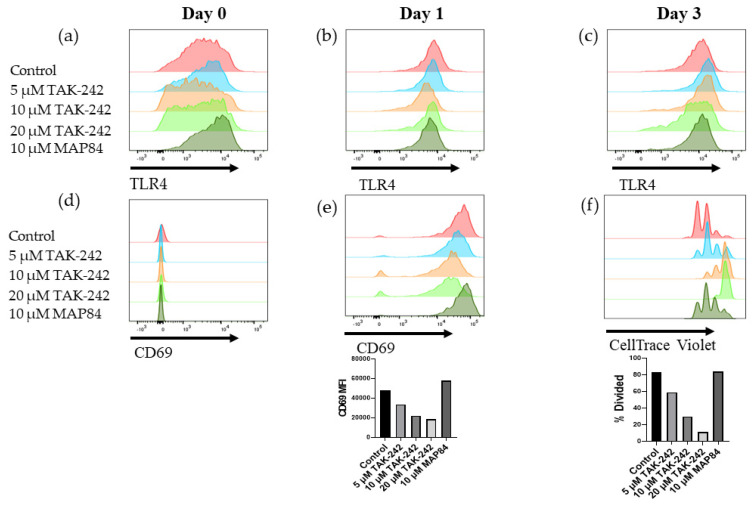
TLR4 expression on CD8+ T cells undergoing polyclonal CD3/CD28 stimulation (**a**) at the beginning of the experimental period, (**b**) after 24 h, and (**c**) after 3 days. CD69 expression was monitored on CD8+ T cells (**d**) at the beginning of the experiment and (**e**) 1 day after beginning the experiment. (**f**) Proliferation was monitored 3 days after stimulating CD8+ T cells with CD3/CD28-coated beads.

**Figure 6 cells-13-00416-f006:**
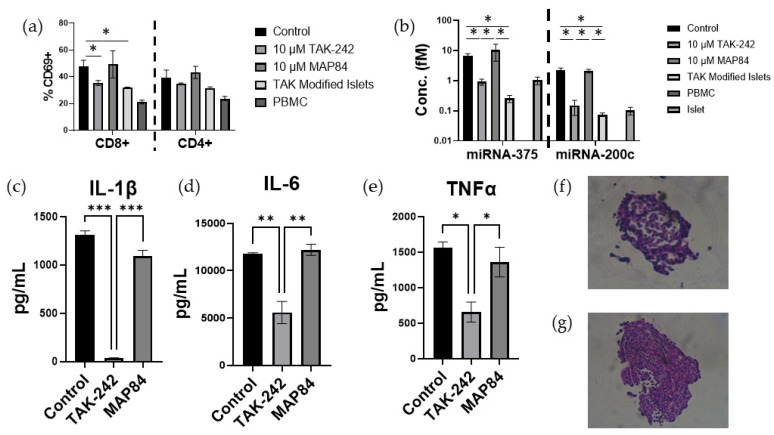
Activation of (**a**) CD8+ and CD4+ T cells 24 h after incubation with allogenic islets. Concentrations of (**b**) miR-375 and miR-200c in the supernatant 24 h after the mixture of allogenic islets and PBMCs. Expression of (**c**) IL-1β, (**d**) IL-6, and (**e**) TNFα after 24 h of incubation of PBMCs and islets. Representative pictures of (**f**) control mixed culture and (**g**) TAK-242 mixed culture. * *p* < 0.05, ** *p* < 0.005, *** *p* < 0.0005.

**Figure 7 cells-13-00416-f007:**
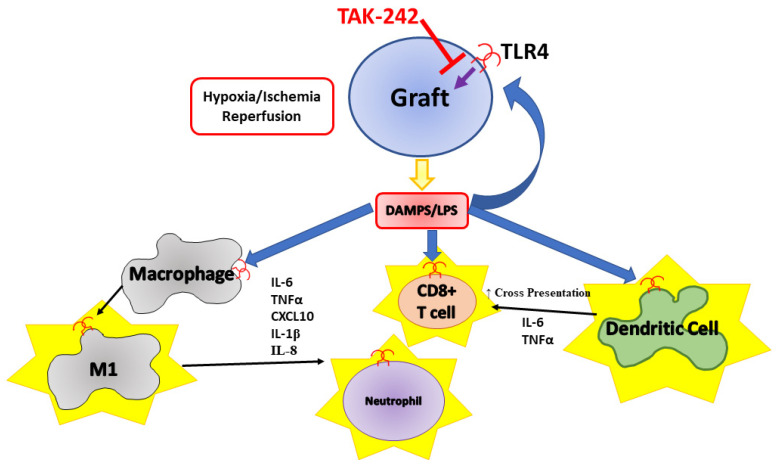
TLR4 plays a significant role in the initiation of innate immune responses that can result in inflammation and destruction of transplanted islet tissue.

**Table 1 cells-13-00416-t001:** Patient and pancreas characteristics at the time of TPIAT as well as islet isolation outcomes.

Variable	Value (*n* = 10)
Gender (female:male)	8:2
Age (years)	40 ± 11.7
Height (cm)	165 ± 9.88
Weight (kg)	70.2 ± 11.82
Body mass index (kg/m^2^)	25.75 ± 3.71
Disease duration (years)	9.4 ± 5.9
Fasting blood glucose (mg/dL)	91.1 ± 12.3
Stimulated blood glucose (mg/dL)	153.5 ± 54.5
Basal C-peptide (ng/mL)	1.91 ± 1.39
Stimulated C-peptide (ng/mL)	7.39 ± 4.48
∆ C-peptide (ng/mL)	5.48 ± 3.47
Initial trimmed pancreas weight (g)	134.6 ± 25.3
Pancreas weight processed (g)	88.0 ± 23.3
Total islet yield (IEQ)	560,973 ± 125,830
Islet particle number (IN)	335,700 ± 96,743
Islet yield (IEQ/g pancreas)	5986 ± 2012
Dose (IEQ/kg patient)	8147 ± 2040

## Data Availability

The data generated in this study are saved in secure institutionally approved databases. The data are available for sharing upon request and approval from the institutional administration.

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
