# Peer review of "Inhibition of Toll-like Receptor 4 Using Small Molecule, TAK-242, Protects Islets from Innate Immune Responses"

_cells, 2024, doi:10.3390/cells13050416_

Round 1
Reviewer 1 Report
Comments and Suggestions for Authors
The manuscript by Mattke et al. presents an interesting strategy to mitigate innate immune inflammatory response to transplanted islets. Accordingly, the authors present data to support the use of TRL4 antagonist, TAK-242, on the surface of islets to mitigate activation of immune response. Particularly the authors present data on suppression of DC activation and activation and proliferation of T cells. Using in vitro assays, the authors show that displaying TAK-242 on surface of islets reduced islet damage with reduction in miR-375 and miR-200c, biomarkers for islet damage. This also correlated with reduction of IL-1β, an inflammatory cytokine. However, the authors do not provide any data on effect of TAK-242 on other innate immune cells including monocytes and neutrophils. Studies suggests that complement activation, infiltration of islets with neutrophils and macrophages plays role in early loss of islets due to IBMIR. The reviewer has following concerns to be addressed to improve the impact of the manuscript.
1. What is the effect of TAK-242 on activation of neutrophils, macrophages and complement activation?
2. Does displaying TAK-242 on islet surface prevent complement activation and infiltration of islets with neutrophils and macrophages?
3. Histological evidence to suggest reduction of IBMIR mediated injury will be of interest to support the efficacy of TAK-242.
4. Preclinical in vivo data will substantially support the idea of TAK-242 mitigating loss of islets to IBMIR.
5. It is not clear if TAK-242 by itself has any effects on islets as it is recognized that islets express TLR4.

Author Response
We thank the reviewer for all of the suggestions, comments, and concerns.
Please see the attached word document for comprehensive responses to all of the comments.

Reviewer 2 Report
Comments and Suggestions for Authors
Abstract:
1- Consider restructuring the second sentence for better clarity. Perhaps: "Innate immune responses, particularly the instant blood-mediated inflammatory reaction and activation of monocytes, play a major role..."
2- The sentence "We then investigate the cellular mechanisms behind this damage using in vitro models of cellular transplantation and how targeting TLR4 can lead to improved outcomes during the peritransplant period" could be shortened to something like: "Using in vitro models, we investigate how TLR4 targeting mitigates cellular damage during the peritransplant period."
3- Briefly mention the potential significance of the study's findings.
Introduction:
1- The introduction could be shortened, particularly the background information on TLR4 and islet transplantation. Focus on the most relevant details and avoid excessive repetition.
2- The flow of the introduction could be improved by reorganizing some information. Consider starting with a broader context of islet transplantation challenges, then introducing TLR4 and its role in innate immunity, followed by the specific hypothesis and research objectives.
3- Some sentences could be clarified for improved readability. For example, "We assess the islet damage using the biomarkers miR-375 and miR-200c based on in vivo and in vitro analyses" could be rephrased as "Our study uses miR-375 and miR-200c as biomarkers to assess islet damage in both animal models and cell culture experiments."
4- Start with a sentence or two summarizing the challenge of islet transplantation and the significance of overcoming it.
Methods:
1- Explain how the sample size of 10 patients was determined for clinical data analysis. Was it based on power calculations or previous studies?
2- While previous studies are mentioned, consider including additional data in this study to definitively validate these miRNAs as reliable markers of islet damage.
3- Provide more information on the chosen concentration of TAK-242 (10 μM) for IBMIR simulation and rationale behind its selection.
4- In addition to miR-375 and miR-200c, include functional assays to directly measure islet viability and insulin secretion following TAK-242 treatment.
5- Consider including experiments that assess the long-term effects of TAK-242 on islet survival and function, perhaps using animal models.
6- Explore the specific mechanisms by which TAK-242 inhibits TLR4 signaling and how it modulates immune cell responses in vitro and in vivo.
7- The authors could definitely consider using other markers for flow cytometry in their study. Including markers like CD45RA and CCR7 to differentiate between naive, effector, and memory T cell populations could reveal if TAK-242 affects specific memory T cell subsets that contribute to islet rejection.
Results:
The results presented in the article are commendable, demonstrating a thorough analysis of the data. However, to enhance the clarity and visual appeal of the findings, I suggest refining the quality and visibility of the bar charts. Improving the resolution and ensuring consistency in labeling and formatting will further elevate the presentation of the results.
Discussion:
Discuss more about the strengths and limitations of MiR-375 and miR-200c biomarkers for assessing islet damage.
Discuss limitations such as sample size, in vitro vs. in vivo models, and potential off-target effects of TAK-242.
Consider adding a figure summarizing the proposed mechanisms of action of TAK-242, which could enhance clarity and visual appeal.
Comments on the Quality of English LanguageMinor editing needed.
Author Response
We greatly appreciate all of the feedback received from this reviewer, and we hope we have addressed all reviewer comments in a satisfactory manner.
Please see the attached word document for a comprehensive list of responses and revisions made to our manuscript.

Round 2
Reviewer 1 Report
Comments and Suggestions for Authors
The reviewer is thankful for the response provided, which makes the manuscript much clearer and understandable. However, a few changes have been suggested to be made to the manuscript.
1. The authors should add a section in the introduction to provide context on the previous study (ref #25) performed in subrenal islet transplantation by targeting TLR4 by using the similar technique that will release TLR4 antagonist in localized fashion. This will allow the reader to follow up on the study performed and also understand the technology behind the study.
2. While the authors stated that TAK-242 potentially inhibits activation of macrophages and neutrophils as suggested by reduced levels of inflammatory cytokines and chemokines. However, the study was based on DC and whole PBMCs. It would be better to cite some published studies in the discussion section that will support that TAK-242 suppress activation of macrophages and neutrophils.
3. In the response section and the discussion the authors has stated "Our results have shown that TAK-242 does not inhibit the coagulation (Fig. 1c)" This does not appear to be the right figure as figure 1c shows the co-relation between miR375 and insulin secretion.
Author Response
We would like to thank the reviewer for their helpful suggestions.
Please see the attachment for comprehensive responses addressing these suggestions.

Reviewer 2 Report
Comments and Suggestions for Authors
Dear Authors,
Thank you for your diligent revisions to the manuscript. I am pleased to recommend its publication.
Best Regards,
Reviewer
Author Response
We thank this reviewer for their time and suggestions. We are grateful for the reviewers approval of our manuscript.